# Choosing Wisely and Learning Deeply: Selective Cross-Modality Distillation via CLIP for Domain Generalization

**Jixuan Leng**                                                           *jleng3@u.rochester.edu*
*University of Rochester*

**Yijiang Li**                                                                    *yli556@jhu.edu*
*Johns Hopkins University*

**Haohan Wang**                                                       *haohanw@illinois.edu*
*University of Illinois Urbana-Champaign*

**Reviewed on OpenReview:** *https://openreview.net/forum?id=4KLwep6mA1*

## Abstract

Domain Generalization (DG), a crucial research area, seeks to train models across multiple domains and test them on unseen ones. In this paper, we introduce a novel approach, namely, Selective Cross-Modality Distillation for Domain Generalization (SCMD). SCMD leverages the capabilities of large vision-language models, specifically CLIP, to train a more efficient model, ensuring it acquires robust generalization capabilities across unseen domains. Our primary contribution is a unique selection framework strategically designed to identify hard-to-learn samples for distillation. In parallel, we introduce a novel cross-modality module that seamlessly combines the projected features of the student model with the text embeddings from CLIP, ensuring the alignment of similarity distributions. We assess SCMD's performance on various benchmarks, where it empowers a ResNet50 to deliver state-of-the-art performance, surpassing existing domain generalization methods. Furthermore, we provide a theoretical analysis of our selection strategy, offering deeper insight into its effectiveness and potential in the field of DG.

## 1 Introduction

Imagine a young pianist learning to perform a complex piece of music. Initially, she listens to her experienced piano teacher's rendition, absorbing the nuances and techniques. Yet, the most crucial learning comes from recording and listening to her own performances. When she listens to her own recordings, she can identify where her performance differs from the perfect melody she has in mind - the ideal, real-world performance. These differences are more valuable for her improvement than simply comparing her performance to the ideal. This is because these disparities directly reflect her unique challenges, and addressing them (with the teacher's help) will have the most direct impact on improving her performance.

This learning scenario is akin to the process of Knowledge Distillation (Hinton et al., 2015). The simpler problems that the pianist initially tackles resemble the easily learnable features of a dataset, which a student model can grasp without assistance. The complex music pieces, however, symbolize the hard-to-grasp parts of the data, where the student model benefits immensely from the teacher model's guidance.

As the pianist matures artistically, she discerns that mastering known pieces alone does not capture the full essence of her musical journey. Wanting to elevate her compositions, she begins to intertwine lyrics into her melodies. This is not a mere juxtaposition of words and tunes; it is a realization that lyrics can heighten the clarity and expressiveness of her music. In the same vein, the harmonization of visual content and natural language in CLIP Radford et al. (2021) is not just for the sake of fusion, but because language can significantly augment the nuances of visual recognition.

Deep learning models, much like the pianists, may falter when faced with unfamiliar terrains or "genres" in the form of out-of-domain data, despite their proficiency in various tasks. This situation frequently arises in real-world applications. Numerous algorithms have been developed to ensure consistency across distributions (Ben-David et al., 2010; Ben-David et al., 2006) and regularize the models to learn domain-invariant features (Lu et al., 2022; Huang et al., 2020; 2022), but they often yield only modest improvements (Gulrajani & Lopez-Paz, 2021) over traditional Empirical Risk Minimization (ERM) technique (Vapnik, 1998).

Inspired by the pianist's pursuit of harmony between melody and lyrics, and her introspective approach to identify discrepancies in her performance to perfect her craft, our work similarly seeks to focus on challenging aspects of training data and incorporate semantic information to better capture the domain-invariant features. In this paper, we present the Selective Cross-Modality Distillation (SCMD) framework.

Rather than relying on the soft target distribution from the teacher model, SCMD emphasizes the discrepancies, specifically, the gap between the student's performance and real-world expectations. Just as the pianist hones in on the variations between her rendition and the ideal composition, SCMD selects hard-to-learn samples in the training data, targeting those for knowledge distillation. This approach, we believe, not only enhances the learning process but also equips the student model with robust feature representations, crucial for navigating unfamiliar terrains.

We have chosen to utilize CLIP as a key component of our approach, not only for its ability to combine visual and linguistic information but also for its proficiency in matching images with textual descriptions. This special capacity of CLIP enhances our framework, offering a more comprehensive knowledge base from which our student models can extract and learn.

**Contributions:**

- We present the Selective Cross-Modality Distillation (SCMD) framework, an innovative departure from traditional methods, emphasizing adaptive sample treatments over the uniform approaches commonly found in current knowledge distillation techniques.

- We emphasize the importance of complex and hard-to-learn training samples and provide a comprehensive theoretical foundation for our selection strategies.

- We propose a cross-modality distillation module within our SCMD framework to leverage the unique capabilities of CLIP, seamlessly integrating linguistic comprehension with visual perception for a more nuanced learning paradigm.

- We substantiate the effectiveness of our SCMD method through empirical evaluations, demonstrating its superior performance and capability to establish new standards on various benchmarks.

## 2 Related Work

### 2.1 Domain Generalization

Domain Generalization (DG) (Muandet et al., 2013) has recently emerged as a key area in machine learning, focused on training models with data from multiple source distributions for application on a distinct target distribution.

DG is generally classified into two facets: data augmentation and invariance regularization (Liu et al., 2023). Data augmentation thread (Shankar et al., 2018; Yue et al., 2019; Gong et al., 2019; Zhou et al., 2020; Huang et al., 2021; Wang et al., 2022b) refines the inputs to facilitate learning of generalizable representations. Invariance regularization techniques (Li et al., 2018b;c; Wang et al., 2017; Akuzawa et al., 2020; Ding et al., 2022; Han et al., 2021; Wang et al., 2022a; Meng et al., 2022; Lee et al., 2022; Ge et al., 2021; Lu et al., 2022) aim for domain-invariant feature representations to enhance generalization. In addition, some methods employ learning strategies like meta-learning (Li et al., 2018a) to simulate domain shifts during training, improving the model's adaptability to unseen domains. Recent studies indicate that weight averaging can further boost DG task performance (Cha et al., 2021; Arpit et al., 2022), contributing to more stable out-of-domain test results.

Several previous studies have leveraged vision-language models to enhance Domain Generalization (DG) performance, closely aligning with our contribution. For instance, Domain Prompt Learning (DPL) (Zhang et al., 2021) employs a lightweight prompt adapter to automatically generate a prompt estimating domain-specific features from unlabeled examples across distributions. However, these generated prompts often lack clear semantic meanings, potentially limiting their effectiveness in certain contexts (Zhou et al., 2022). Other research (Li et al., 2022) dispatches appropriate pretrained models, including CLIP, to each sample based on their generalization ability. Another approach (Cha et al., 2022) reformulates the DG objective using mutual information with oracle models, including CLIP.

Recent work (Huang et al., 2023) introduces RISE, a method that utilizes both absolute and relative distances for distilling CLIP for domain generalization. Building on this foundation, our approach prioritizes the identification and selection of hard-to-learn samples for knowledge distillation. This combination of ideas, though seemingly straightforward, represents a new direction that has not been extensively explored in prior research.

## 2.2 Contrastive Language-Image Pre-Training

CLIP (Radford et al., 2021), a vision-language model utilizing contrastive loss to align visual and text encoders within a shared feature space, has recently garnered significant attention.

Pretrained on 400 million image-text pairs (Radford et al., 2021), CLIP effectively aligns semantic meanings between images and sentences, demonstrating its promise for generic visual representation learning and zero-shot transfer via prompt (Radford et al., 2021; Jia et al., 2021; Yang et al., 2022b;a; Yao et al., 2022; You et al., 2022). Notably, CLIP matches a fully-supervised ResNet101 model's performance with a 76.2% top-1 accuracy rate on the ImageNet (Deng et al., 2009) validation set and even outperforms it on the ImageNet Sketch Dataset with 60.2% accuracy rate.

These results highlight CLIP's exceptional capabilities in tasks like image classification. More importantly, CLIP's extensive pretraining not only bolsters its standalone performance but also demonstrates its potential to transfer extensive knowledge to other architectures, offering new possibilities for enhancing model generalization across diverse domains.

In this paper, we present a novel method to distill knowledge from CLIP, a multi-modal vision-language model, into a single-modal student model. By transitioning from multi-modal to single-modal distillation, we aim to enhance the student model's domain generalization, opening up new avenues for leveraging these potent models.

## 2.3 Knowledge Distillation

Knowledge distillation, introduced by Hinton et al. (Hinton et al., 2015), is a pivotal technique for balancing model performance and computational complexity by training a smaller student network with the soft output of a larger teacher model. This approach has spurred extensive research in model compression (Cheng et al., 2017) and knowledge transfer (Tan et al., 2018).

Numerous distillation techniques have emerged, including feature-based knowledge transfer methods (Romero et al., 2015a; Bengio et al., 2013; Zagoruyko & Komodakis, 2017; Kim et al., 2018a; Heo et al., 2019a) that align embeddings from certain layers, (Zhang et al., 2018) that train teacher and student models concurrently. The design of loss functions operating on the outputs of both models has also been a significant research area, with notable methods including $l_1$ (Kim et al., 2018b), $l_2$ (Chen et al., 2020; Passban et al., 2021; Wang et al., 2020), Maximum Mean Discrepancy (MMD) (Huang & Wang, 2017), KL divergence (Chen et al., 2018; Passalis et al., 2020a;b), and cross-entropy losses (Xu et al., 2020; Liu et al., 2019).

However, most studies focus on homologous-architecture distillation, leaving cross-architecture distillation relatively untapped. Recently, Liu et al. (Liu et al., 2022) made significant progress in this area by mapping a CNN's feature space into a transformer's attention and feature space using a partially cross-attention projector and a group-wise linear projector. With a cross-view robust training scheme, they achieved remarkable performance on ImageNet (Deng et al., 2009).

Our approach diverges from conventional methods, introducing fresh perspectives on knowledge distillation through two key innovations. Firstly, we implement a novel selection mechanism that identifies hard-to-learn samples, which enhances the student model's depth of understanding. Secondly, we leverage CLIP's multi-modal capabilities by employing a cross-modality module. This strategy not only facilitates a profound transfer of both visual and linguistic knowledge but also significantly enhances the domain generalization capability of the student model.

## 3  Methods

In this section, we provide a detailed description of our Selective Cross-Modality Distillation Framework, which leverages a pretrained CLIP model (Radford et al., 2021) with fixed weights as the guiding teacher model.

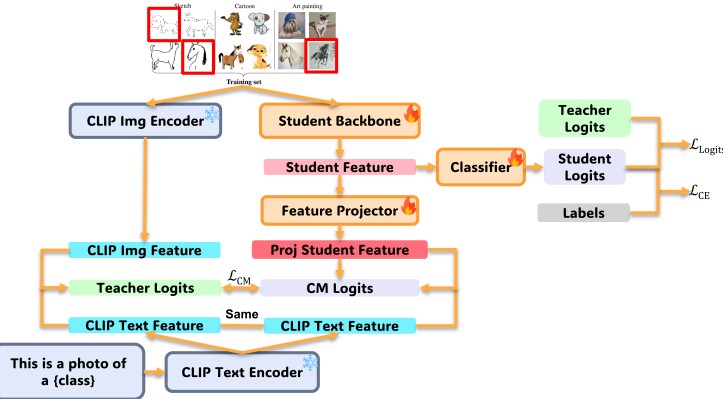

Figure 1: SCMD that features a selection mechanism to focus on hard-to-learn samples and a cross-modality module that projects the student's feature into CLIP multi-modal space for alignment.

### 3.1  Vanilla Knowledge Distillation

The vanilla knowledge distillation process is

$$
\begin{aligned}
\hat{\theta}_{\mathrm{KD}} = \arg\min_{\theta} \sum_{(\mathbf{x}_i, \mathbf{y}_i) \in (\mathbf{X}, \mathbf{Y})} & \mathcal{L}(\phi(\mathbf{x}_i), f(\mathbf{x}_i; \theta)) \\
& + \lambda_1 \mathcal{H}(f(\mathbf{x}_i; \theta), \mathbf{y}) \\
& + \lambda_2 R(\phi, f, p, q, \mathbf{x}_i, \mathbf{y}_i)
\end{aligned}
\tag{1}
$$

where $\phi$ is the teacher model, $f(\cdot; \theta)$ is the student model. $(\mathbf{X}, \mathbf{Y})$ is the standard dataset that is used for training the student model. $\mathcal{L}$ is a generic distance function (can be the KL divergence between soft target distributions), $\mathcal{H}$ represents a generic loss function (usually cross-entropy loss), and $R$ is an arbitrary designed regularization. $p$ and $q$ correspond to certain layers or attention maps.

### 3.2  Selection Mechanism

$$
S = \mathbf{x}_i : \mathbf{x}_i \in \mathbf{X}, i \in I \quad \text{where } I = i : \mathcal{H}(f(\mathbf{x}_i), \mathbf{y}_i) \geq \tau
\tag{2}
$$

In the preceding equation, $\mathbf{X}$ denotes the batch of samples, $\mathbf{x}_i$ an individual sample, and $\mathbf{y}_i$ its true label. The set $S$ consists of selected samples. The function $\mathcal{H}(f(\mathbf{x}_i; \theta), \mathbf{y}_i)$ computes the cross-entropy loss for the $i$-th sample, while $I$ contains indices of samples in the batch with a cross-entropy loss exceeding the threshold $\tau$.

Cross-entropy loss quantifies the divergence between the predicted probability distribution and the actual labels. High cross-entropy indicates challenging samples or model uncertainty. In our optimization, we use this loss to identify hard-to-learn samples. During each forward pass, samples with higher losses are selected for adjustment, a methodology also adopted in prior works (Huang et al., 2022; Byrd & Lipton, 2019; Chang et al., 2017; Katharopoulos & Fleuret, 2018).

The uniqueness of our method lies not in the recognition of hard-to-learn samples but in its integration within knowledge distillation. By doing so, we harness the teacher model's rich knowledge more efficiently, optimizing the learning of the student model. We explore the theoretical foundation of our selection mechanism in Section 4.

The "hard-to-learn" tag for the samples can change each iteration. However, to ensure holistic learning across the entire training dataset, we switch to full-batch training for the final $k\%$ of the training epochs.

### 3.3 Cross-Modality Module

Various feature-based knowledge distillations have been explored (Romero et al., 2015a; Bengio et al., 2013; Zagoruyko & Komodakis, 2017; Kim et al., 2018a; Heo et al., 2019a); however, direct alignment of classification features often presents challenges. To address this, we exploit the robust cross-modal alignment capabilities of the CLIP (Radford et al., 2021) and employ a cross-modality distillation strategy.

In the context of our method, the student features are transformed into the CLIP's (Radford et al., 2021) multi-modal space using a linear projection. This is analogous to how CLIP projects its image features into a multi-modal space to achieve alignment with text embeddings. This transformation bridges the semantic gap between the student model and the teacher model, facilitating a more effective knowledge-transfer process. After projection, we calculate the scaled pairwise cosine similarity with the text embeddings derived from the CLIP (Radford et al., 2021) model.

Our cross-modality loss is expressed as follows:

$$\mathcal{L}_{CM} = D_{\mathrm{KL}}(p^t||(p^s)')$$
$$\text{where } p^t \text{ is the soft target distribution of CLIP and} \tag{3}$$
$$(p^s)' = \sigma(\gamma \cdot W(e(\mathbf{x}_i; \theta_e)) \cdot \phi_{\text{text}}; T = t)$$

In this equation, $\gamma$ is a scale factor, which adjusts the magnitude of the projected student feature. $e$ represents the backbone of the student model. A linear projection $W$ is applied to the student feature $e(x_i; \theta_e)$, and $\phi_{\text{text}}$ represents the text embedding of CLIP. $\sigma$ is the softmax function parameterized by the distillation temperature $T$.

In order to generate unbiased text features using CLIP, we use a generic template: "this is a photo of a {class}". This method helps us avoid incorporating any human prior knowledge about the dataset, ensuring that the feature generation process remains objective and is not influenced by any preconceived human understanding of the data.

During the inference phase, we omit the feature projector and rely solely on the student model's backbone and its associated classifier for generating predictions, introducing no additional computation overhead.

### 3.4 SCMD

Figure 1 illustrates the overall framework of our proposed method.

The final training objective can be summarized as follows:

$$
\begin{aligned}
\hat{\theta}_{\text{SCMD}} = \arg\min_{\theta} \sum_{(\mathbf{x}_i, \mathbf{y}_i) \in (\mathbf{X}, \mathbf{Y})} & \lambda_1 \mathcal{H}(f(\mathbf{x}_i; \theta), \mathbf{y}_i) \\
& + \lambda_2 \mathcal{L}_{\text{logits}} + \lambda_3 \mathcal{L}_{\text{CM}} \\
\text{where } & \mathcal{L}_{\text{logits}} = D_{\text{KL}}(p^t || p^s) \\
& \text{and } \mathbf{x}_i \text{ and } \mathbf{y}_i
\end{aligned}
\tag{4}
$$

are the selected samples and corresponding labels

## 4    Theoretical Evidence for Selection Strategy

To be consistent with the notation, we let $(\mathbf{X}, \mathbf{Y})$ denote the standard data set and $(\mathbf{x}_i, \mathbf{y}_i)$ be one of the samples. We let $\boldsymbol{P}$ denote a distribution and $\mathcal{P}$ denote the distribution of distributions. We let $f(\cdot; \theta)$ denote the student model, $\phi$ denote the teacher model, and $r(\cdot)$ denote the risk. For convenience of notation, we allow $r(\cdot)$ to be parameterized by a distribution or by a dataset.

When $r(\cdot)$ is parameterized by a dataset, we have the empirical risk as

$$
r\big((\mathbf{X}, \mathbf{Y})\big) = \frac{1}{n} \sum_{(\mathbf{x}_i, \mathbf{y}_i) \in (\mathbf{X}, \mathbf{Y})} \mathcal{L}(f(\mathbf{x}_i; \theta), \mathbf{y}_i),
$$

where $\mathcal{L}$ is a generic loss function.

When $r(\cdot)$ is parameterized by a distribution, we have the expected risk as

$$
r\big(\boldsymbol{P}\big) = \mathbb{E}_{(\mathbf{x}_i, \mathbf{y}_i) \sim \boldsymbol{P}} \mathcal{L}(f(\mathbf{x}_i; \theta), \mathbf{y}_i),
$$

For simplicity of discussion, we only use $r\big(\boldsymbol{P}, \epsilon\big)$ to denote robustness performance when we do not need to specify how the test distribution deviates from the training distribution.

**Assumption 1.** *for any data pair $(\mathbf{x}_i, \mathbf{y}_i)$ studied in the context of this paper, there is a gold standard labeling function (albeit unknown) that $\mathbf{y}_i = f(\mathbf{x}_i)$.*

We believe this assumption is fundamental for numerous works studying the robustness behaviors of models with respect to feature perturbations, especially in the context of OOD robustness, where the test dataset is manually collected rather than generated by noise addition. Intuitively, this assumption stipulates that a musical piece recognized in the training phase must also be identified as the same piece in the testing phase, despite substantial shifts in the performance style or instrument used. In other words, these variations in representation, akin to distribution shifts in data, should not alter the fundamental recognition of the piece, preserving the semantics of the data.

**Lemma 4.1.** *Given Assumptions A1 such that there is a gold standard labeling function for source and target domains. For two arbitrary distributions $\boldsymbol{P}'$ and $\boldsymbol{P}$,*

$$
r(\boldsymbol{P}') \leq r(\boldsymbol{P}) + tv(\boldsymbol{P}', \boldsymbol{P})
$$

*where $tv$ denotes the total variation.*

**Proof.** We leave the proof in Appendix A

**Lemma 4.2.** *Given the assumption that samples are independent and identically distributed, hypothesis space $\Theta$ and any $\delta > 0$, with probability at least $1 - \delta$, we have*

$$
r\big(\boldsymbol{P}'\big) \leq r\big((\mathbf{X}, \mathbf{Y})_{\boldsymbol{P}}\big) + tv(\boldsymbol{P}', \boldsymbol{P}) + \xi(n_{(\mathbf{X}, \mathbf{Y})_{\boldsymbol{P}}}, \Theta, \delta)
$$

*where we let $n_{(\mathbf{X}, \mathbf{Y})_{\boldsymbol{P}}}$ denote the number of sample sizes in the finite dataset $(\mathbf{X}, \mathbf{Y})_{\boldsymbol{P}}$, $\xi$ is a vanilla term that connects the number of samples and hypothesis space with generalization error bound.*

**Proof.** We leave the proof in Appendix A

The above results demonstrate that empirical robustness is determined by three factors: the divergence between training and test distributions, the measurable empirical error on the training distribution, and a technical term influenced by sample size and hypothesis space. Therefore, the critical term that will bound the robustness performance is how the training distribution deviates from the testing distribution. This intuitively gives us the idea that training with the distributions that are the most similar to the test distribution will benefit the model most.

The above results apply to arbitrary distributions $\boldsymbol{P} \sim \mathcal{P}$. However, this does not necessarily encode the characteristics of the cases we are studying: some samples are hard for the model to learn.

To address this, we consider datasets generated by multiple distributions, some of which present more challenging learning scenarios. We represent these as a set $P$, consisting of m distributions, i.e., $P = \{\boldsymbol{P}_1, \boldsymbol{P}_2, \ldots, \boldsymbol{P}_m\}$. Each data point is considered to be sampled from these distributions. For the convenience of discussion, we use $tv(\boldsymbol{P}', P)$ to denote the average divergence between the distributions within the set. $tv(\boldsymbol{P}', P) := \sum_i^m tv(\boldsymbol{P}', \boldsymbol{P}_i)/m, \forall \boldsymbol{P}_i \in P$.

Finally, we use $s()$ to denote the distribution selection mechanism, and compare two selection mechanisms: selecting the hard-to-learn samples (denoted as $s_1$) and selecting random samples (denoted as $s_2$).

**Lemma 4.3.** *$\mathcal{P}$ is continuous and has a finite expected value; for the two selection mechanism that are formally defined as*

$$tv(s_1(P), P) = \sup_{\boldsymbol{P} \in P} tv(\boldsymbol{P}, P), \quad \mathbb{E}_{\mathcal{P}} tv(s_2(P), P) = 0$$

*for a fixed testing dataset $\boldsymbol{P}'$, with the assumption that $tv(P, \boldsymbol{P}') = tv(P, \boldsymbol{P}) + tv(\boldsymbol{P}, \boldsymbol{P}'), \forall \boldsymbol{P} \in P$ we have*

$$\mathbb{E}_{\mathcal{P}}\Big[tv(s_1(P), \boldsymbol{P}')\Big] \leq \mathbb{E}_{\mathcal{P}}\Big[tv(s_2(P), \boldsymbol{P}')\Big]$$

**Proof.** We leave the proof in Appendix A

Our result compares the upper-bounded differences between the two training distribution selection strategies, and our results suggest that selecting the hard-to-learn samples will lead to a tighter generalization error bound.

Another important factor to note is that, given assumption A1 and Lemma 4.1 and 4.2, the selection strategy applicable to our theoretical discussion (i.e. $tv(s_1(P), P) = \sup_{\boldsymbol{P} \in P} tv(\boldsymbol{P}, P)$) is only when selecting the hard-to-learn samples according to the label of the samples (thus cross-entropy loss). Other selection strategies such as selecting based on KL-divergence or distillation loss (experimented in Section 6.2) despite might following a similar goal, does not strictly match our theoretical discussion with will likely lead to an error bound in between $s_1$ and $s_2$. Therefore, with the support of the theoretical discussion, we argue that the most effective hard-to-learn selection mechanism is to be based on cross-entropy loss.

Another possible question is that the assumption $tv(P, \boldsymbol{P}') = tv(P, \boldsymbol{P}) + tv(\boldsymbol{P}, \boldsymbol{P}'), \forall \boldsymbol{P} \in P$ might appear strong. In fact, the proof will hold with any assumptions that describe the concept that the more different one distribution is from the average of the training set, the more it will benefit the testing distribution. In the typical domain generalization setting, where there are no guaranteed connections between training and testing distributions, we believe this is one of the practical assumptions we can consider, also widely used in practical by other domain generalization literature (Huang et al., 2022; Byrd & Lipton, 2019; Chang et al., 2017; Katharopoulos & Fleuret, 2018).

## 5 Experiment

In this section, we demonstrate the effectiveness of our proposed method using the DomainBed (Gulrajani & Lopez-Paz, 2021) benchmark and compare it to the current state-of-the-art DG techniques.

## 5.1 Experimental Setup

We adhere to the protocol set out in (Gulrajani & Lopez-Paz, 2021) for our experimental setup and assess the performance of SCMD using VLCS (Fang et al., 2013), PACS (Li et al., 2017), OfficeHome (Venkateswara et al., 2017), TerraIncognita (Beery et al., 2018), and DomainNet (Peng et al., 2019). It is noteworthy that CLIP does not perform well on the TerraIncognita (Beery et al., 2018) dataset, and we will explore these results in the discussion section.

Due to the intensive computing requirements of DomainBed's (Gulrajani & Lopez-Paz, 2021) hyperparameter search protocol, we take a more simplified approach. We restrict our research to five distinct hyperparameter combinations, each tested three times. We assign 80% of the data for training and 20% for validation, choose the model based on the training-domain validation performance, and report the results on the held-out domain.

To ensure a fair comparison with other methods, we employ ResNet50 (He et al., 2016) pretrained on ImageNet1k (Deng et al., 2009) as the student model and CLIP ViT-B/32 (Radford et al., 2021) as the teacher model, aligning our approach with existing research in the field.

Consistent with findings from previous studies (Cha et al., 2021; Arpit et al., 2022), we also incorporate weight averaging into our experiments to access SCMD performance. This technique has been shown to mitigate the discrepancy between training-domain validation performance and out-of-domain test performance.

Eight RTX 3090 GPUs are utilized for all experiments.

## 5.2 Experimental results

| Algorithm | Ens MA | VLCS | PACS | OffHome | TerraInc | DNet | Avg |
|---|---|---|---|---|---|---|---|
| Teacher (CLIP with ViT-B/32) | No | $78.4 \pm 0.0$ | $94.7 \pm 0.1$ | $79.6 \pm 0.1$ | $19.0 \pm 0.1$ | $54.0 \pm 0.0$ | 65.1 |
| ERM (Vapnik, 1998) | No | $77.5 \pm 0.4$ | $85.5 \pm 0.2$ | $66.5 \pm 0.3$ | $46.1 \pm 1.8$ | $40.9 \pm 0.1$ | 63.3 |
| CORAL (Sun & Saenko, 2016) | No | $78.8 \pm 0.6$ | $86.2 \pm 0.3$ | $68.7 \pm 0.3$ | $47.6 \pm 1.0$ | $41.5 \pm 0.1$ | 64.6 |
| VREx (Krueger et al., 2021) | No | $78.3 \pm 0.2$ | $84.9 \pm 0.6$ | $66.4 \pm 0.6$ | $46.4 \pm 0.6$ | $33.6 \pm 2.9$ | 61.9 |
| RSC (Huang et al., 2020) | No | $77.1 \pm 0.5$ | $85.2 \pm 0.9$ | $65.5 \pm 0.9$ | $46.6 \pm 1.0$ | $38.9 \pm 0.5$ | 62.7 |
| ERM + SWAD (Cha et al., 2021) | Yes | $79.1 \pm 0.1$ | $88.1 \pm 0.1$ | $70.6 \pm 0.2$ | $50.0 \pm 0.3$ | $46.5 \pm 0.1$ | 66.9 |
| CORAL + SWAD (Cha et al., 2021) | Yes | $78.9 \pm 0.1$ | $88.3 \pm 0.1$ | $71.3 \pm 0.1$ | $51.0 \pm 0.1$ | $46.8 \pm 0.0$ | 67.3 |
| AdaClust (Thomas et al., 2021) | No | $78.9 \pm 0.6$ | $87.0 \pm 0.3$ | $67.7 \pm 0.5$ | $48.1 \pm 0.1$ | $43.3 \pm 0.5$ | 64.9 |
| MIRO + SWAD (Cha et al., 2022) | Yes | $\underline{79.6 \pm 0.2}$ | $88.4 \pm 0.1$ | $72.4 \pm 0.1$ | $\mathbf{52.9 \pm 0.2}$ | $47.0 \pm 0.0$ | 68.1 |
| EoA (Arpit et al., 2022) | Yes | 79.1 | 88.6 | 72.5 | $\underline{52.3}$ | 47.4 | 68.0 |
| Model rata(Greedy)(Ramé et al., 2022) | Yes | $78.7 \pm 0.2$ | $\mathbf{90.5 \pm 0.2}$ | $73.4 \pm 0.3$ | $49.2 \pm 0.9$ | $47.7 \pm 0.0$ | 67.9 |
| Model rata(Uni)(Ramé et al., 2022) | Yes | 78.3 | 89.8 | $\underline{73.5}$ | 52.0 | $\underline{47.7}$ | $\underline{68.3}$ |
| SCMD (ours) | Yes | $\mathbf{80.9 \pm 0.2}$ | $\underline{90.1 \pm 0.0}$ | $\mathbf{74.8 \pm 0.1}$ | $51.3 \pm 0.2$ | $\mathbf{48.4 \pm 0.0}$ | $\mathbf{69.1}$ |

Table 1: Performance benchmarking on 5 datasets of the DomainBed benchmark. The gray background shows our proposed method. Experiments report the performance based on training-domain validation accuracy follow (Gulrajani & Lopez-Paz, 2021). 'Ens/MA' stands for ensemble/moving average. (best in **bold** and second underlined)

Table 1 shows that our proposed method achieves the best performance on the DomainBed benchmark. It outperforms existing methods on all datasets, with Model Ratatouille (Ramé et al., 2022) coming in second.

Model Ratatouille (Ramé et al., 2022) utilizes a technique that adjusts the model on multiple extra tasks to obtain different weight initializations. These weights are then adjusted for the desired tasks, and the final model is created by taking the average of these weights. This is exemplified by Model Ratatouille (Uniform), which averages a total of 60 models to achieve the final result, as shown in the Table.

In contrast, our proposed method uses a single teacher model and evaluates performance using just one student model. Furthermore, our method is orthogonal to existing DG methods, potentially providing additional avenues and perspectives in the broader landscape of DG research.

# 6 Ablation Studies

We conduct a thorough analysis of SCMD by breaking it down into its components and examining each using the PACS dataset. To evaluate the effectiveness of the proposed cross-modality module and the selection mechanism, we follow the same standardized hyperparameter search protocol as the main experience, ensuring consistency and comparability.

## 6.1 Impact of the Cross-Modality Module

Table 2 (Top Section) presents the comprehensive results of our method alongside its different variations.

- **"Vanilla KD"** (Hinton et al., 2015) denotes the conventional knowledge distillation technique where the KL divergence between the predicted distributions of the student and teacher models is minimized.

- **"SCMD (logits)"** is the combination of the selection mechanism and the minimization of KL divergence.

- **"SCMD (logits + CM)"** represents the full version of our method, including all our proposed components.

We have included weight averaging into the Vanilla KD to guarantee a fair comparison and show the effectiveness of our proposed components.

As shown in the top part of Table 2, our selection mechanism alone leads to a 0.5% improvement in comparison to the average performance of Vanilla KD. Additionally, our cross-modality (CM) module further boosts the performance by 0.6%. When both are combined, our proposed methodology offers a significant increase in performance, surpassing Vanilla KD by a total of 1.1%. These results demonstrate the combined power and effectiveness of our proposed approach.

| Algorithm | Avg |
|---|---|
| Vanilla KD (Hinton et al., 2015) | $89.0 \pm 0.3$ |
| SCMD (logits) | $89.5 \pm 0.2$ |
| SCMD (logits + CM) (full method) | $\mathbf{90.1 \pm 0.0}$ |
| SCMD (no selection) | $89.4 \pm 0.3$ |
| SCMD (selection based on KL) | $89.6 \pm 0.3$ |
| SCMD (selection based on distill loss) | $89.8 \pm 0.2$ |
| SCMD (selection based on focal loss) | $89.2 \pm 0.4$ |
| SCMD (selection based on CE loss) | $\mathbf{90.1 \pm 0.0}$ |

Table 2: Performance evaluations: (Top) Impact of the cross-modality module in SCMD on the PACS dataset. (Bottom) SCMD performance with different strategies for selecting hard-to-learn samples on the PACS dataset. (best in **bold**)

## 6.2 Empirical Validation of Our Theoretical Analysis

As illustrated in Table 2 (Bottom Section), we employed various selection strategies for the samples.

- **"no selection"** represents the baseline scenario where the entire training dataset is used without any hard-to-learn sample selection.

- **"selection based on KL"** refers to sample selection based on the KL-divergence between the predicted distributions of the student and teacher models.

- **"selection based on distill loss"** implies that samples are chosen according to the distill loss, as defined in Eq 4.

- **"selection based on focal loss"** implies that samples are chosen according to the focal loss (Lin et al., 2017).

- **"selection based on CE loss"** denotes our proposed selection strategy.

It is evident that any selection strategy except "selection based on focal loss" yields better results than the "no selection" baseline. Our proposed "selection based on CE loss" approach is the most successful on the PACS dataset, outperforming "selection based on KL" by 0.5%, "selection based on distill loss" by 0.3%, and the no selection strategy by 0.7%. It is worth noting that the "distill loss" (Eq 4) includes the cross-entropy loss, which could explain why its performance is similar to "selection based on CE loss", albeit slightly lower.

These results provide empirical support to our theoretical proposition: "Other selection strategies such as selecting based on KL-divergence or distillation loss despite might following a similar goal, do not strictly match our theoretical discussion which will likely lead to an error bound between $s_1$ and $s_2$." Therefore, with the support of the theoretical discussion and empirical evidence, we argue that the most effective hard-to-learn selection mechanism is to be based on cross-entropy loss.

# 7 Discussion and Limitations

## 7.1 Impact of Prompts Variations

In order to reduce bias in the feature extraction process with CLIP, we use a template that does not contain any human-derived insights, which is: "this is a photo of a {}". This template anchors the feature generation process in a way that is not dependent on any particular domain, thus avoiding the impact of any human preconceptions.

Our experiments show that the prompt "photo" was the most effective for optimizing performance. We also found that slight changes to the prompt, such as "a photo of a {}" and "this is a photo of a {}", had little effect on the success of the distillation process. This demonstrates the resilience of feature distillation to minor changes in the prompt structure. Table 3 provides further details on the ablation studies.

| Prompt | Avg |
|---|---|
| this is an art of a {} | $88.9 \pm 0.4$ |
| this is a sketch of a {} | $89.6 \pm 0.2$ |
| this is a cartoon of a {} | $88.7 \pm 0.4$ |
| a photo of a {} | $89.9 \pm 0.2$ |
| this is a photo of a {} | $89.9 \pm 0.4$ |
| a {} | $88.8 \pm 0.2$ |

Table 3: The performance of SCMD on PACS with different prompts, using the same hyper-parameters for three trials.

## 7.2 Analysis on TerraInc Performance

CLIP itself exhibits sub-optimal zero-shot performance, which becomes evident when we distill it into ResNet50 using the TerraIncognita dataset, as detailed in Table 1. Despite this challenge, our approach still offers advantages; notably, SCMD-no-KD outperforms the baseline ERM method. This suggests that pre-conditioning the CLIP model through fine-tuning before distillation could be a beneficial strategy to enhance performance for similar tasks.

Table 4: Impact of KD on TerraIncognita. 'MA': Moving Average. 'SCMD_no_KD': variant when KD not used.

| Algorithm | MA | Terra Avg |
|---|---|---|
| ERM | No | $46.1 \pm 1.8$ |
| SCMD (without KD) | Yes | $53.1 \pm 0.5$ |
| SCMD | Yes | $51.3 \pm 0.2$ |

## 7.3 Experiments on various student models

We investigate the effect of different teacher models and model sizes by applying SCMD to the PACS dataset. We use ResNet152 and ResNet18 and follow the same experimental setup and hyperparameter search protocol as in our previous experiments.

Table 2 demonstrates that our SCMD approach consistently outperforms Vanilla KD across various scenarios, including when different vision encoders from CLIP, such as RN101, are used as the teacher model. Specifically, SCMD achieved an improvement of approximately 0.6% over Vanilla KD. This result highlights the versatility and effectiveness of our method in both cross-architecture and homologous-architecture distillation scenarios.

Our approach demonstrates significant enhancements in performance, yielding a noteworthy improvement of 3.4% over the ERM technique and 0.8% over Vanilla KD when using ResNet152 as the student model.

| Algorithm | CLIP | Student | Avg |
|-----------|------|---------|-----|
| ERM(reproduced) | ViT-B/32 | / | $94.7 \pm 0.1$ |
| ERM(reproduced) | RN101 | / | $94.9 \pm 0.1$ |
| Vanilla KD | RN101 | RN-50 | $87.9 \pm 0.3$ |
| SCMD | | | $\mathbf{88.5 \pm 0.2}$ |
| Vanilla KD | ViT-B/16 | RN-50 | $89.2 \pm 0.3$ |
| SCMD | | | $\mathbf{89.5 \pm 0.3}$ |
| ERM (reproduced) | ViT-B/32 | RN-152 | $88.7 \pm 0.5$ |
| Vanilla KD | | | $91.3 \pm 0.1$ |
| SCMD | | | $\mathbf{92.1 \pm 0.2}$ |
| ERM (Ye et al., 2021) | ViT-B/32 | RN-18 | $81.5 \pm 0.0$ |
| RSC (Huang et al., 2020) | | | $82.8 \pm 0.4$ |
| IRM (Arjovsky et al., 2019) | | | $81.1 \pm 0.3$ |
| MMD (Li et al., 2018b) | | | $81.7 \pm 0.2$ |
| Vanilla KD | | | $84.8 \pm 0.3$ |
| SCMD | | | $\mathbf{85.0 \pm 0.2}$ |

Figure 2: Evaluation of SCMD's performance across various student and CLIP model architectures on the PACS dataset

Even with a smaller model like ResNet18, our method maintains strong performance relative to other DG methods, showing a marginal improvement of 0.2% over Vanilla KD. This slight difference may be attributed to the substantial capacity gap between ResNet18 and CLIP.

## 8 Conclusion

In this paper, we introduce Selective Cross-Modality Distillation (SCMD) for Domain Generalization, a novel approach that enhances the traditional knowledge distillation framework. Our method is designed to supplement existing DG techniques. We introduce a cross-modality module that leverages the robust cross-modal alignment capabilities of CLIP. Central to SCMD is a selection mechanism that is both theoretically grounded and empirically validated through extensive experimentation. Our results demonstrate the efficacy of our proposed method, underscoring its potential to advance domain generalization.

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

## Appendix

## A    Theoretical Evidence for Selection Strategy

To be consistent with notation, we let $(\mathbf{X}, \mathbf{Y})$ denote the standard dataset, and $(\mathbf{x}_i, \mathbf{y}_i)$ as one of the samples. We let $\boldsymbol{P}$ denote a distribution and $\mathcal{P}$ denote the distribution of distributions. We let $f(\cdot; \theta)$ denote the student model, $\phi$ denote the teacher model, and $r(\cdot)$ denote the risk. For the convenience of notations, we allow $r(\cdot)$ to be parameterized by a distribution or by a dataset.

**Lemma A.1.** *Given assumptions A1 such that there is a gold standard labeling function for source and target domains. For two arbitrary distributions $\boldsymbol{P}'$ and $\boldsymbol{P}$,*

$$r(\boldsymbol{P}') \leq r(\boldsymbol{P}) + tv(\boldsymbol{P}', \boldsymbol{P})$$

*where tv denotes the total variation.*

*Proof.* Recall that we are assuming the **same labelling function**, let $\sigma$ and $\sigma'$ be the density functions of $\boldsymbol{P}$ and $\boldsymbol{P}'$

$$r(\boldsymbol{P}') = r(\boldsymbol{P}') + r(\boldsymbol{P}) - r(\boldsymbol{P}) \leq r(\boldsymbol{P}) + \mid r(\boldsymbol{P}') - r(\boldsymbol{P}) \mid$$
$$\leq r(\boldsymbol{P}) + \int \mid \sigma(\mathbf{x}) - \sigma'(\mathbf{x}) \mid \mid f(\mathbf{x}; \theta) - \mathbf{y} \mid d\mathbf{x}$$
$$\leq r(\boldsymbol{P}) + tv(\boldsymbol{P}', \boldsymbol{P})$$

$\square$

**Lemma A.2.** *Given assumption that samples are independent and identically distributed, hypothesis space $\Theta$ and any $\delta > 0$, with probability at least $1 - \delta$, we have*

$$r(\boldsymbol{P}') \leq r((\mathbf{X}, \mathbf{Y})_{\boldsymbol{P}}) + tv(\boldsymbol{P}', \boldsymbol{P}) + \xi(n_{(\mathbf{X}, \mathbf{Y})_{\boldsymbol{P}}}, \Theta, \delta)$$

*where we let $n_{(\mathbf{X}, \mathbf{Y})_{\boldsymbol{P}}}$ denote the number of sample sizes in the finite dataset $(\mathbf{X}, \mathbf{Y})_{\boldsymbol{P}}$, $\xi$ is a vanilla term that connects the number of samples and hypothesis space with generalization error bound.*

*Proof.* Recall that we are assuming that samples are independent and identically distributed, we have $\xi(n_{(\mathbf{X}, \mathbf{Y})_{\boldsymbol{P}}}, \Theta, \delta) = 2\mathcal{R}(\mathcal{L}) + \sqrt{(\log 1/\delta)/2n}$ where $\mathcal{R}(\mathcal{L})$ stands for Rademacher complexity and $\mathcal{L} = \{l_\theta \mid \theta \in \Theta\}$ where $l_\theta$ is the loss function corresponding to the student model $f(\cdot; \theta)$

$$r(\boldsymbol{P}') \leq r(\boldsymbol{P}) + tv(\boldsymbol{P}', \boldsymbol{P})$$
$$\leq r((\mathbf{X}, \mathbf{Y})_{\boldsymbol{P}}) + tv(\boldsymbol{P}', \boldsymbol{P}) + \xi(n_{(\mathbf{X}, \mathbf{Y})_{\boldsymbol{P}}}, \Theta, \delta)$$

This is an direct application of Lemma A.1 and generalization error bound.    $\square$

The above results demonstrate that empirical robustness is determined by three factors: the divergence between training and test distributions, the measurable empirical error on the training distribution, and a technical term influenced by sample size and hypothesis space. Therefore, the critical term that will bound the robustness performance is how the training distribution deviates from the testing distribution. This intuitively gives us the idea that training with the distributions that are the most similar to the test distribution will benefit the model most.

The above results apply to arbitrary distributions $\boldsymbol{P} \sim \mathcal{P}$. However, this does not necessarily encode the characteristics of the cases we are studying: some samples are hard for the model to learn.

To address this, we consider datasets generated by multiple distributions, some of which present more challenging learning scenarios. We represent these as a set $P$, consisting of m distributions, i.e., $P = \{\boldsymbol{P}_1, \boldsymbol{P}_2, \ldots, \boldsymbol{P}_m\}$. Each data point is considered as sampled from these distributions. For the convenience

of discussion, we use $tv(\boldsymbol{P}', P)$ to denote the average divergence between the distributions within the set. $tv(\boldsymbol{P}', P) := \sum_i^m tv(\boldsymbol{P}', \boldsymbol{P}_i)/m, \quad \forall \boldsymbol{P}_i \in P.$

Finally, we use $s()$ to denote the distribution selection mechanism, and we compare two selection mechanisms: selecting the hard-to-learn samples (denoted as $s_1$) and selecting random samples (denoted as $s_2$).

**Lemma A.3.** $\mathcal{P}$ *is continuous and has a finite expected value; for the two selection mechanism that are formally defined as*

$$tv(s_1(P), P) = \sup_{\boldsymbol{P} \in P} tv(\boldsymbol{P}, P), \quad \mathbb{E}_{\mathcal{P}} tv(s_2(P), P) = 0$$

*for a fixed testing dataset $\boldsymbol{P}'$, with the assumption that $tv(P, \boldsymbol{P}') = tv(P, \boldsymbol{P}) + tv(\boldsymbol{P}, \boldsymbol{P}'), \forall \boldsymbol{P} \in P$ we have*

$$\mathbb{E}_{\mathcal{P}}\Big[tv(s_1(P), \boldsymbol{P}')\Big] \leq \mathbb{E}_{\mathcal{P}}\Big[tv(s_2(P), \boldsymbol{P}')\Big]$$

*Proof.* Based on our definition of $s_1$ and $s_2$,

$$\mathbb{E}_{\mathcal{P}} tv(s_2(P), \boldsymbol{P}') = \mathbb{E}_{\mathcal{P}} tv(P, \boldsymbol{P}')$$

And based on our assumption that $tv(P, \boldsymbol{P}') = tv(P, \boldsymbol{P}) + tv(\boldsymbol{P}, \boldsymbol{P}')$, we have

$$\mathbb{E}_{\mathcal{P}}\Big[tv(s_1(P), \boldsymbol{P}') - tv(s_2(P), \boldsymbol{P}')\Big]$$
$$= \mathbb{E}_{\mathcal{P}}\Big[\inf_{\boldsymbol{P} \in P} tv(\boldsymbol{P}, \boldsymbol{P}') - \mathbb{E}_{\mathcal{P}} tv(P, \boldsymbol{P}')\Big] \leq 0$$

$\square$

Our result compares the upper-bounded differences between the two training distribution selection strategies, and our results suggest that selecting the hard-to-learn samples will lead to a tighter generalization error bound.

Another important factor to note is that, given assumption A1 and Theorem 0.1, the selection strategy applicable to our theoretical discussion (i.e. $tv(s_1(P), P) = \sup_{\boldsymbol{P} \in P} tv(\boldsymbol{P}, P)$) is only when selecting the hard-to-learn samples according to the label of the samples (thus, cross-entropy loss). Other selection strategies such as selecting based on KL-divergence or distillation loss despite might following a similar goal, do not strictly match our theoretical discussion, which will likely lead to an error bound in between $s_1$ and $s_2$. Therefore, with the support of the theoretical discussion, we argue that the most effective hard-to-learn selection mechanism is to be based on cross-entropy loss.

Another possible question is that Assumption $tv(P, \boldsymbol{P}') = tv(P, \boldsymbol{P}) + tv(\boldsymbol{P}, \boldsymbol{P}')$ might appear strong. In fact, the proof will hold with any assumptions that describe the concept that the more different one distribution is from the average of the training set, the more it will benefit the testing distribution. In the typical domain generalization setting, where there are no guaranteed connections between training and testing distributions, we believe this is one of the practical assumptions we can consider, also widely used in practical context by other domain generalization literature (Huang et al., 2022; Byrd & Lipton, 2019; Chang et al., 2017; Katharopoulos & Fleuret, 2018)

# B Method

## B.1 Algorithm

## B.2 Selection Mechanism

$$S = \mathbf{x}_i : \mathbf{x}_i \in \mathbf{X}, i \in I \quad \text{where } I = i : \mathcal{H}(f(\mathbf{x}_i), \mathbf{y}_i) \geq \tau \tag{5}$$

In the preceding equation, $\mathbf{X}$ denotes the batch of samples, $\mathbf{x}_i$ an individual sample, and $\mathbf{y}_i$ its true label. The set $S$ consists of selected samples. The function $\mathcal{H}(f(\mathbf{x}_i; \theta), \mathbf{y}_i)$ computes the cross-entropy loss for the $i$-th sample, while $I$ contains indices of samples in the batch with a cross-entropy loss exceeding the threshold $\tau$.

---

**Algorithm 1** Selective Cross-Modality Distillation

---

**Input**: Dataset $(\mathcal{X}, \mathcal{Y})$ of size $n$; Percentile of hard-to-learn samples per batch $\rho$; Percentile of full-batch training $\kappa$; Batch size $\eta$; Maximum number of iterations $T$; Feature Projector $P$; pretrained teacher model $\phi$; Student model $\theta$ (randomly initialized)

**Output**: Trained student model $\theta$

> **while** t ≤ T **do**
>> **while** t ≤ $(1 - \kappa)T$ **do**
>>> Identify top $\rho$ percentile samples with highest Cross-entropy loss based on Eq 2
>>> For selected samples, compute student features and project to CLIP's multi-modal space via $P$.
>>> Distill knowledge from the teacher model $\phi$ to the student model $\theta$ using Eq 4
>> **end while**
>> Distill knowledge from $\phi$ to $\theta$ across the entire batch and calculate the final loss with Eq 4
> **end while**
> **return** optimized student model $\theta$

---

## B.3 Cross-Modality Module

$$\mathcal{L}_{CM} = D_{\mathrm{KL}}(p^t || (p^s)')$$

where $p^t$ is the soft target distribution of CLIP and $\qquad\qquad$ (6)

$$(p^s)' = \sigma(\gamma \cdot W(e(\mathbf{x}_i; \theta_e)) \cdot \phi_{\text{text}}; T = t)$$

In this equation, $\gamma$ is a scale factor, which adjusts the magnitude of the projected student feature. $e$ represents the backbone of the student model. A linear projection $W$ is applied to the student feature $e(x_i; \theta_e)$, and $\phi_{\text{text}}$ represents the text embedding of CLIP. $\sigma$ is the softmax function parameterized by the distillation temperature $T$.

## B.4 SCMD

$$\hat{\theta}_{\mathrm{SCMD}} = \arg\min_{\theta} \sum_{(\mathbf{x}_i, \mathbf{y}_i) \in (\mathbf{X}, \mathbf{Y})} \lambda_1 \mathcal{H}(f(\mathbf{x}_i; \theta), \mathbf{y}_i)$$
$$+ \lambda_2 \mathcal{L}_{\text{logits}} + \lambda_3 \mathcal{L}_{\mathrm{CM}}$$
$$\text{where } \mathcal{L}_{\text{logits}} = D_{\mathrm{KL}}(p^t || p^s) \qquad\qquad (7)$$
$$\text{and } \mathbf{x}_i \text{ and } \mathbf{y}_i$$

are the selected samples and their corresponding labels

# C More Analysis

## C.1 Parameter Sensitivity Analysis

To reiterate the definition of the parameters for clarity: for implementation purposes, we set $\tau$ to be the percentile of samples selected as hard-to-learn, with $\tau = 0.25$ indicating the selection of samples with a loss exceeding the 75th percentile. Meanwhile, $k$ denotes the fraction of steps before the transition to full-batch training occurs; for example $k = 0.25$ implies that this shift happens during the last 25% of the steps.

The table presents the performance of our model across various domains of PACS as we adjust the values of $k$ and $\tau$, keeping all other parameters constant during the experiment.

$k$ **Sensitivity (with $\tau = 0.2$):** The model's performance remains relatively stable across an array of $k$ values, as evidenced by the average accuracy consistently around 90.0%.

$\tau$ **Sensitivity (with $k = 0.25$):** Having very few samples is not helpful to the learning process. And a slight decrement in the model's performance is observed as the value of $\tau$ increases, after reaching 0.2. This trend can be ascribed to the dilution of the impact of hard-to-learn samples as their proportion increases.

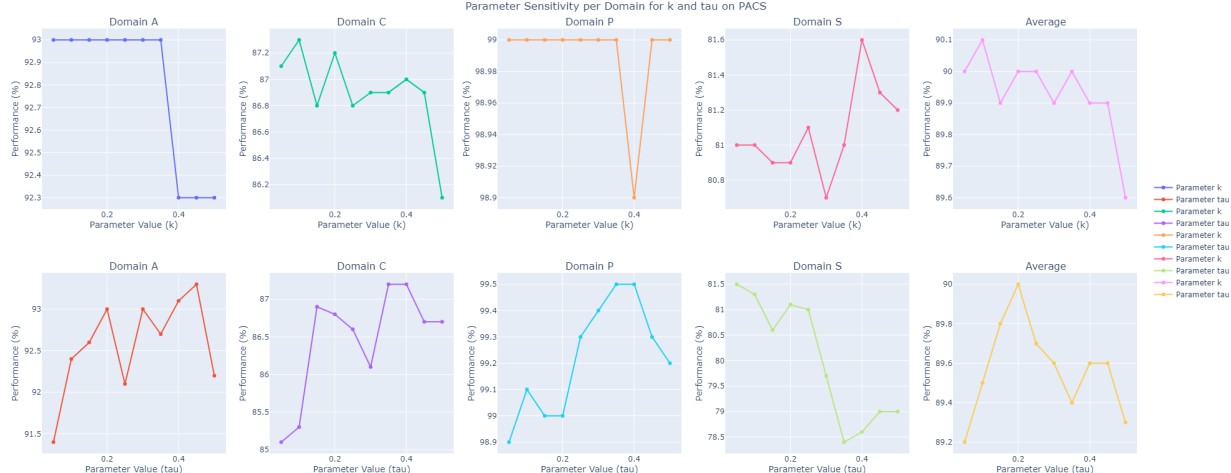

Figure 3: Sensitivity analysis for $\tau$ and $k$

Specifically, as more samples are classified as hard-to-learn (with a higher $\tau$), the advantage of focusing on these challenging samples is somewhat mitigated, subtly influencing the overall model performance.

## C.2  Are high-loss samples outliers?

| Exclude Top-K | Domain A | Domain C | Domain P | Domain S | Average |
|---|---|---|---|---|---|
| 0 | 93.0 | 86.8 | 99.0 | 81.1 | 90.0 |
| 1 | 92.5 | 86.8 | 99.2 | 80.8 | 89.8 |
| 2 | 92.1 | 86.0 | 99.3 | 80.3 | 89.4 |
| 3 | 93.0 | 86.4 | 99.3 | 80.6 | 89.8 |
| 4 | 92.6 | 86.4 | 98.8 | 80.8 | 89.6 |
| 5 | 91.7 | 85.2 | 99.1 | 79.7 | 88.9 |

Table 5: Performance upon the exclusion of top hard-to-learn samples

As depicted in Table 5, our empirical analysis confirms the critical role of the highest-loss samples in influencing the model's overall performance. The data reveals a clear pattern: excluding these high-loss samples consistently leads to a reduction in performance metrics. Specifically, the deliberate removal of the top 1 and top 2 highest-loss samples results in a quantifiable decrease in model efficacy, with average performance dropping from 90.0% to 89.8%, and then to 89.4%. These findings highlight that high-loss samples are not merely outliers but are crucial for the model's learning process, significantly enhancing its ability to generalize and perform accurately across various domains.

## C.3  Training Time Analysis

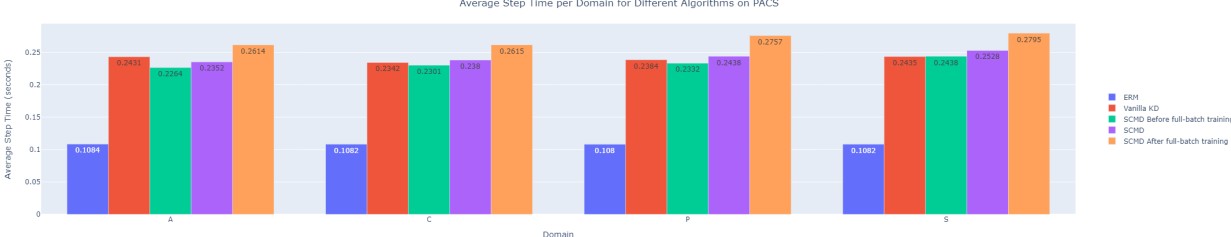

Figure 4: Average step time per domain for different algorithms on PACS

The results indicate that the average training overhead of SCMD is comparable to that of Vanilla KD. This is attributed to our training strategy that focuses on hard-to-learn samples for the majority of the duration, transitioning to full-batch training only in the last $k$ steps. This method balances the overall training overhead.

### C.4 Compare with other KD methods

| Algorithm | MA | Avg |
|-----------|-----|------|
| FitNet (Romero et al., 2015b) | Yes | $88.4 \pm 0.2$ |
| BSS (Heo et al., 2019b) | Yes | $89.3 \pm 0.1$ |
| RKD (Park et al., 2019) | Yes | $87.4 \pm 0.2$ |
| Vanilla (Hinton et al., 2015) | Yes | $89.0 \pm 0.3$ |
| **SCMD** | Yes | $90.1 \pm 0.0$ |

Table 6: SCMD vs. other KD on PACS. "MA": Moving Average

At the core of SCMD is the knowledge distillation process. To evaluate the effectiveness of SCMD, we conduct comparative experiments with other knowledge distillation methods on the PACS dataset. The results in Table 6 demonstrate that SCMD outperforms these contemporary techniques.

## D Full Results

Full details of the results from Table 1 in the main paper are provided in Table 9.

### D.1 hyperparameters search space

We adhere to the experimental setup described in the DomainBed (Gulrajani & Lopez-Paz, 2021) paper. The specifics of our setup are outlined below:

- **Data Split:** We partition datasets into 80% training and 20% validation sets. Model selections are based on training domain validation performances, and we report on the corresponding test domain.

- **hyperparameters:** Although many hyperparameters follow (Gulrajani & Lopez-Paz, 2021), deviations are documented in Table 7.

- **Batch & Decay:** We adjust our batch size and weight decay following the guidelines of (Cha et al., 2021).

- **Dropout:** The ResNet dropout rate is set to 0 to mitigate excessive randomness.

- **Learning Rate:** We abandon the rate of $1 \times 10^{-5}$ because it converges too slowly, and instead focus on rates of $3 \times 10^{-5}$ and $5 \times 10^{-5}$.

Table 8 outlines the search space specific to our algorithm's hyperparameters. We consistently set $\lambda_1$, the balance factor for cross-entropy, to 1, while conducting random sweeps for the other weight factors.

| Parameter | Default | **DomainBed** (Gulrajani & Lopez-Paz, 2021) | **SWAD** (Cha et al., 2021) | **Ours** |
|-----------|---------|--------------------------------|------------------|--------|
| batchsize | 32 | $2^{U(3,5.5)}$ | 32 | 32 |
| learning rate | 5e-5 | $10^{U(-5,-3.5)}$ | [1e-5, 3e-5, 5e-5] | [3e-5, 5e-5] |
| ResNet dropout | 0 | [0.0, 0.1, 0.5] | [0.0, 0.1, 0.5] | 0 |
| weight decay | 0 | $10^{U(-6,-2)}$ | [1e-4, 1e-6] | [1e-4, 1e-6] |

Table 7: Unconditional hyperparameter search space. (U and list indicate Uniform distribution and random choice, respectively)

| Parameter | Default Value | Sweep range |
|---|---|---|
| $\lambda_1$ (for CE loss) | 1 | 1 |
| $\lambda_2$ | 0.5 | $U(0.5, 1.0)$ |
| $\lambda_3$ | 0.5 | $U(0.5, 1.0)$ |
| last_k_epoch | 0.25 | $U(0.2, 0.4)$ |
| hard-to-learn sample percentage | 1/3 | $[0.2, 0.25, 0.3]$ |
| temperature | 3.0 | $U(2.0, 5.0)$ |

Table 8: Algorithm-specific hyperparameter search space. (U and list indicate Uniform distribution and random choice, respectively)

| | | | | | | | | |
|---|---|---|---|---|---|---|---|---|
| **VLCS** | | | | | | | | |
| | **CLIP** | **Student** | **A** | **C** | **P** | **S** | | **Avg** |
| SCMD | ViT-B/32 | RN50 | $98.8 \pm 0.1$ | $64.6 \pm 0.4$ | $78.2 \pm 0.3$ | $81.9 \pm 0.3$ | | $80.9 \pm 0.2$ |
| **PACS** | | | | | | | | |
| | | | **C** | **L** | **S** | **V** | | **Avg** |
| SCMD | ViT-B/32 | RN50 | $92.9 \pm 0.3$ | $86.0 \pm 0.3$ | $99.0 \pm 0.1$ | $82.3 \pm 0.1$ | | $90.1 \pm 0.0$ |
| SCMD | ViT-B/32 | RN152 | $94.0 \pm 0.2$ | $89.1 \pm 0.5$ | $99.3 \pm 0.1$ | $85.9 \pm 0.6$ | | $92.1 \pm 0.2$ |
| SCMD | ViT-B/32 | RN18 | $84.7 \pm 0.1$ | $80.7 \pm 0.5$ | $96.2 \pm 0.1$ | $78.6 \pm 0.6$ | | $85.0 \pm 0.2$ |
| SCMD | RN101 | RN50 | $90.2 \pm 0.8$ | $83.4 \pm 0.3$ | $99.1 \pm 0.1$ | $81.1 \pm 0.8$ | | $88.5 \pm 0.2$ |
| Vanilla KD | ViT-B/32 | RN50 | $91.2 \pm 0.2$ | $85.2 \pm 0.3$ | $99.4 \pm 0.1$ | $80.3 \pm 0.9$ | | $89.0 \pm 0.3$ |
| Vanilla KD | ViT-B/32 | RN152 | $93.8 \pm 0.1$ | $87.3 \pm 0.4$ | $99.6 \pm 0.0$ | $84.7 \pm 0.2$ | | $91.3 \pm 0.1$ |
| Vanilla KD | ViT-B/32 | RN18 | $84.1 \pm 0.5$ | $80.0 \pm 0.2$ | $96.7 \pm 0.2$ | $78.5 \pm 0.6$ | | $84.8 \pm 0.3$ |
| Vanilla KD | RN101 | RN50 | $89.8 \pm 0.2$ | $80.8 \pm 0.9$ | $99.1 \pm 0.1$ | $81.8 \pm 0.5$ | | $87.9 \pm 0.3$ |
| **OfficeHome** | | | | | | | | |
| | | | **A** | **C** | **P** | **R** | | **Avg** |
| SCMD | ViT-B/32 | RN50 | $72.7 \pm 0.0$ | $58.7 \pm 0.1$ | $82.4 \pm 0.1$ | $85.5 \pm 0.1$ | | $74.8 \pm 0.1$ |
| **TerraIncognita** | | | | | | | | |
| | | | **L100** | **L38** | **L43** | **L46** | | **Avg** |
| SCMD | ViT-B/32 | RN50 | $52.9 \pm 1.1$ | $45.3 \pm 0.1$ | $60.5 \pm 0.5$ | $46.6 \pm 1.0$ | | $51.3 \pm 0.2$ |

| | | | | | | | | | |
|---|---|---|---|---|---|---|---|---|---|
| **DomainNet** | | | | | | | | | |
| | | | **clip** | **info** | **paint** | **quick** | **real** | **sketch** | **Avg** |
| SCMD | ViT-B/32 | RN50 | $66.2 \pm 0.1$ | $25.2 \pm 0.0$ | $56.9 \pm 0.2$ | $14.3 \pm 0.2$ | $70.4 \pm 0.0$ | $57.4 \pm 0.1$ | $48.4 \pm 0.0$ |

Table 9: Detailed Performance of SCMD and Vanilla KD

