# OpenReview forum: "Choosing Wisely and Learning Deeply: Selective Cross-Modality Distillation via CLIP for Domain Generalization"
_TMLR — Accepted by TMLR_

### Review · Reviewer_v6HT · 2024-01-26

**Summary Of Contributions:**

The paper proposes Selective Cross-Modality Distillation for Domain Generalization (SCMD) to train an image encoder that generalizes well to unseen domains.
The authors claim their proposed methods—hard-to-learn sample selection and a cross-modality module—excel at training a model for the DomainBed benchmark, which evaluates the domain generalization capability of models.
Their methods differ from existing ones by exploiting a pre-trained CLIP model.
The ablation study showed that each method contributed to the performance boost. The paper also provides assumption-driven theoretical proof explaining why SCMD outperforms other baselines.

**Audience:**

Yes

**Claims And Evidence:**

Yes

**Requested Changes:**

Please clarify what the unique contribution of this submission is, compared to the two papers I mentioned in the previous section.
Given that the additional computational burden of using ViT-B/16 CLIP should be minimal, results using ViT-B/16 CLIP as a teacher model should also be included for a comprehensive analysis.

**Strengths And Weaknesses:**

Although backed by strong assumptions and incalculable metrics (e.g., TV), the authors have attempted to explain why their methods work well.
However, I am doubtful whether this method fairly compares with other baseline methods shown in Table 1, as all other methods do not employ a powerful pre-trained foundational model (i.e., CLIP).
I note that few papers published in ICCV 2023 [1,2], which employed CLIP for the task of domain generalization, did not compare their method with existing DG approaches head-to-head.
Moreover, the performance of these two approaches is remarkably higher than that of the proposed method, with the caveat that they used ViT-B/16 CLIP, whereas this submission used ViT-B/32 CLIP.
Since the submission claims to be about domain generalization methods, I believe there is room for improvement.
The authors should clearly denote why this method is tailored for domain generalization, not feature distillation, as I think the logical steps in this paper could be a better fit for the topic of feature distillation.

[^1]: A Sentence Speaks a Thousand Images: Domain Generalization through Distilling CLIP with Language Guidance
[^2]: PromptStyler: Prompt-driven Style Generation for Source-free Domain Generalization

---

> ### Author Response · Authors · 2024-02-10
> **Response to Reviewer 3**
>
> We sincerely appreciate the reviewer's recognition of our theoretical proof. We have addressed the concerns and implemented the suggested changes as detailed below:
>
> **Q1. Comparison with the suggested papers and ViT-B/16 Results**
>
> We appreciate the reviewer's suggestion to compare our work with [1,2].
>
> Our approach shares a couple of similarities with [1], such as the use of CLIP as a teacher model and application of KD to enhance the performance of a smaller student model, such as ResNet-50. However, there are several distinct aspects and methodological choices that set our work apart and contribute to the differences in performance.
>
> *Teacher Backbone and Computational Overhead*: As highlighted in their Table 6, [1] leverages a mixed teacher (MT), combining ViT-B/16 and RN101, to augment performance. While this approach is innovative, it introduces additional computational overhead due to the utilization of multiple teacher models. In contrast, our method employs a single teacher model for KD. As the following table shows, compared to the single-teacher model performance reported in [1], despite the different teacher backbone, SCMD achieves a performance comparable to [1].
>
> *Handling of low Zero-shot Performance on Terra*:
> The approach in [1] involves fine-tuning CLIP on Terra to address its low zero-shot performance before conducting KD, as detailed in their Section 4.3. This step, while effective, adds complexity and potential biases due to the fine-tuning process on a specific domain. SCMD maintains the integrity of the original CLIP model without domain-specific fine-tuning, offering a more generalized performance.
>
> In line with the reviewer's suggestion, we incorporated ViT-B/16 as the backbone for the teacher model in our experiments. This adaptation yielded a performance of $89.5\%$ on PACS, which is marginally higher than the results reported in [1] without the use of Mixed Teachers (MT).
>
> | Algorithms        | Teacher           | PACS          | VLCS | OfficeHome | Terra | DomainNet |
> |-------------------|-------------------|---------------|------|------------|-------|-----------|
> | SCMD              | ViT-B/32          | 90.1 $\pm$ 0.0| 80.9 $\pm$ 0.2    | 74.8 $\pm$ 0.1        | 51.3 $\pm$ 0.2    | 48.4 $\pm$ 0.0    |
> | SCMD              | ViT-B/16          | 89.5 $\pm$ 0.3| NA                | NA                    | NA                | NA                |
> | RISE              | ViT-B/16          | 89.4          | 81.7              | 71.6                  | 52.3              | NA                |
> | RISE (with MT)    | ViT-B/16  + RN101 | 90.2          | 82.4              | 72.6                  | 54.0              | NA                |
>
> Our research, akin to [1], centers on distilling knowledge from CLIP to enhance the performance of smaller models, a strategy divergent from the approach in [2]. While [2] aims to optimize the CLIP model itself for improved domain generalization, our focus is on leveraging the pre-trained strengths of CLIP through Knowledge Distillation (KD).
>
>
> **Q2. Why Domain Generalization**
>
> We value the reviewer's insights but believe our method aligns more closely with domain generalization for the following reason:
>
> Domain Generalization aims to train models that perform well on unseen data, whereas Feature Distillation aims to transfer intermediate features to student models.
>
> DG not only emphasizes the complexity and applicability required for real-world scenarios, but also establishes a context within which feature distillation can be applied and evaluated. We have included a comparative analysis with traditional Knowledge Distillation (KD) methods like FitNet [3] and RKD [4] in appendix Table 6.
>
> **Q3. Unique Contribution**
>
> We thank the reviewer for highlighting the importance of clearly stating our unique contributions. Our contributions are summarized as follows:
>
> 1. We utilizes CLIP to enhance lightweight student models, similar to [1], but with distinct methods.
>
> 2. Our work diverges from [1] by not focusing on distance losses; instead, prioritizes the identification and selection of hard-to-learn samples for KD.
>
> 3. Our work acts as a bridge between worst-case training scenarios and KD.
>
> 4. Contrasts with [2], which optimizes CLIP for DG, our work demonstrates how to effectively distill CLIP into lightweight models for broader applicability.
>
> [1]: A Sentence Speaks a Thousand Images: Domain Generalization through Distilling CLIP with Language Guidance
>
> [2]: PromptStyler: Prompt-driven Style Generation for Source-free Domain Generalization
>
> [3]: FitNets: Hints for Thin Deep Nets
>
> [4]: Relational Knowledge Distillation

---

> > ### Comment · Reviewer_v6HT · 2024-02-22
> > **comment**
> >
> > I appreciate the comprehensive reply from the authors, which addressed most of my concerns. However, I'm intrigued by the observation that ViT-B/16 distilled SCMD underperformed in PACS compared to ViT-B/32 distilled SCMD, despite the former having a more capable teacher.
> >
> > Given that the authors' response has clarified my questions, I am proceeding to update my recommendations accordingly.

---

### Review · Reviewer_cdPz · 2024-01-27

**Summary Of Contributions:**

This paper addresses the problem of domain generalization. In domain generalization, one wants to learn about data from one domain and achieve good performance on unseen data from a new but related domain. The proposed method selects hard-to-learn samples from the source domain in order to learn better generalization characteristics.

**Audience:**

Yes

**Broader Impact Concerns:**

Domain Generalization has a net positive research impact

**Claims And Evidence:**

Yes

**Requested Changes:**

Change requests:

Please address the questions asked in the strengths & weaknesses section.

Notation: The capital letter P is used for both the linear projection and for the data distribution P_(X,Y) (when in boldface). They are hard to discern.

**Strengths And Weaknesses:**

# Strengths:

* The introduction starts with an analogy to explain the concept of the proposed method. A quote from the analogy is ‘lyrics can heighten the clarity and expressiveness of her music.‘ This also raises a question, however, as introducing lyrics too early can discourage the student with their inability. How does one discern when is the right moment to introduce new complexity?

* The related work gives a strong overview of the literature.

* The paper makes a bridge between the field of hard-to-learn samples and the field of knowledge distillation.

# Weaknesses:

* In Table 1, the improvements are generally less than 1 point percent. Could the authors clarify why they think that the added complexity is worth the small increase in performance?

* The title of the paper is ‘Choosing Wisely’, but the motivation for ‘selecting hard examples’ is missing, except for the high-level analogy in the introduction. If the motivation is to reduce the overall required amount of computation, then the computation time should be reported experimentally. For example, Table 1 could report empirical runtimes. If, on the other hand, the motivation is to select hard examples to improve learning, then I would expect a study on that. For example, how do different selection criteria differ in their influence on the final performance? Or if there is a fraction of hard-to-learn samples, it would be interesting to see how varying that fraction influences results.

* Where should the line be drawn between hard-to-learn samples and plain outliers? Moreover, where should the line be drawn between hard-to-learn samples and incorrectly labeled data? We know from the literature that popular large-scale datasets could contain incorrectly or inconsistently labeled data [1].

* The experimental results could have been stronger with an ablation of the selection criterion. As the title of the paper is ‘Choosing wisely’, it is a natural question to ask how wisely one should choose. Equation 2 introduces a parameter tau that seems to be a threshold about how low the cross-entropy can be. A very instructive experiment could be to sweep different values of tau and study domain generalization performance. If computation is a bottleneck, one could select a fixed number of samples from the cross-entropy ordering (e.g. temperature 1), and interpolate that to random sampling (e.g. temperature 0).

[1] Beyer, Lucas, et al. "Are we done with imagenet?." arXiv preprint arXiv:2006.07159 (2020).

---

> ### Author Response · Authors · 2024-02-10
> **Response to Reviewer 2 (Part 1)**
>
> We sincerely appreciate the reviewer's recognition of how our work connects hard-to-learn samples with knowledge distillation. We have addressed the concerns and implemented the suggested changes as detailed below:
>
> **Q1. Could the authors clarify why they think that the added complexity is worth the small increase in performance?**
>
> We appreciate the reviewer's concern regarding the trade-off between added complexity and the performance boost. As demonstrated in Table 1 of the main paper, the performance gain is significant when measured by standard deviation across datasets. Furthermore, our proposed approach achieves this performance gain with a computational overhead comparable to that of Vanilla KD as shown by the accompanying table.
>
>
> **Q2. motivation for ‘selecting hard examples’ and an ablation of the selection criteria**
>
> We appreciate the reviewer's feedback regarding the motivation behind selecting hard-to-learn samples in our methods.
>
> The primary motivation of our work is to enhance the learning process of the student model and to balance computational overhead. We have conducted an ablation study on different selection criteria, the results of which are presented in Table 2 of the main section. Per requests, we have included the sensitivity analysis and computational time records in the appendix and the accompanying tables. The specifics of these analyses are detailed as follows:
>
> 1. Parameter Sensitivity Analysis:
>
> The table presents the performance of our model across various domains of PACS as we adjust the values of $k$ and $\tau$, maintaining all other parameters constant during the experiment. To reiterate the definition of the parameters for clarity: for implementation purposes, we set $\tau$ to be the percentile of samples selected as hard-to-learn.
>
> *$k$ Sensitivity (with $\tau = 0.2$)*: The model's performance remains relatively stable across an array of $k$ values, as evidenced by the average accuracy consistently around 90.0\%.
>
> *$\tau$ Sensitivity (with $k = 0.25$)*: Having very few samples is not helpful to the learning process. And a slight decrement in the model's performance is observed as the value of $\tau$ increases, after reaching 0.2. This trend can be ascribed to the dilution of the impact of hard-to-learn samples as their proportion increases. Specifically, as more samples are classified as hard-to-learn (with a higher $\tau$), the advantage of focusing on these challenging samples is somewhat mitigated, subtly influencing the overall model performance.
>
> | Parameter | Domain A | Domain C | Domain P | Domain S | Average |
> |-----------|----------|----------|----------|----------|---------|
> | **Parameter $k$ (with $\tau = 0.2$)** | | | | | |
> | 0.05      | 93.0     | 87.1     | 99.0     | 81.0     | 90.0    |
> | 0.1       | 93.0     | 87.3     | 99.0     | 81.0     | 90.1    |
> | 0.15      | 93.0     | 86.8     | 99.0     | 80.9     | 89.9    |
> | 0.2       | 93.0     | 87.2     | 99.0     | 80.9     | 89.9    |
> | 0.25      | 93.0     | 86.8     | 99.0     | 86.9     | 90.0    |
> | 0.3       | 93.0     | 86.9     | 99.0     | 80.7     | 89.9    |
> | 0.35      | 93.0     | 86.9     | 99.0     | 81.0     | 90.0    |
> | 0.4       | 92.3     | 87.0     | 98.9     | 81.6     | 89.9    |
> | 0.45      | 92.3     | 86.9     | 99.0     | 81.3     | 89.9    |
> | 0.5       | 92.3     | 86.1     | 99.0     | 81.2     | 89.6    |
> | **Parameter $\tau$ with ($k = 0.25$)** | | | | | |
> | 0.05      | 91.4     | 85.1     | 98.9     | 81.5     | 89.2    |
> | 0.1       | 92.4     | 85.3     | 99.1     | 81.3     | 89.5    |
> | 0.15      | 92.6     | 86.9     | 99.0     | 80.6     | 89.8    |
> | 0.2       | 93.0     | 86.8     | 99.0     | 81.1     | 90.0    |
> | 0.25      | 92.1     | 86.6     | 99.3     | 81.0     | 89.7    |
> | 0.3       | 93.0     | 86.1     | 99.4     | 79.7     | 89.6    |
> | 0.35      | 92.7     | 87.2     | 99.5     | 78.4     | 89.4    |
> | 0.4       | 93.1     | 87.2     | 99.5     | 78.6     | 89.6    |
> | 0.45      | 93.3     | 86.7     | 99.3     | 79.0     | 89.6    |
> | 0.5       | 92.2     | 86.7     | 99.2     | 79.0     | 89.3    |

---

> > ### Author Response · Authors · 2024-02-10
> > **Response to Reviewer 2 (Part 2)**
> >
> > 2. Computational Efficiency:
> >
> > Computational efficiency is a key aspect of our methodology. This is demonstrated by the added plot in the appendix and the subsequent tables, which illustrate the average training time per step across four domains in the PACS dataset, each domain serving as a test domain in turn. SCMD Before and SCMD After denote the average training time per step for SCMD before and after the transition to full-batch training, respectively, while SCMD represents the overall average.
> >
> > The training overhead of SCMD is comparable to that of Vanilla KD. This is attributed to our training strategy that focuses on hard-to-learn samples for the majority of the duration, transitioning to full-batch training only in the last $k$ steps. This method balances the overall training overhead.
> >
> > | Model         | Domain A | Domain C | Domain P | Domain S |
> > |---------------|----------|----------|----------|----------|
> > | ERM           | 0.1084   | 0.1082   | 0.1080   | 0.1082   |
> > | Vanilla KD    | 0.2431   | 0.2342   | 0.2384   | 0.2435   |
> > | SCMD Before   | 0.2264   | 0.2301   | 0.2332   | 0.2438   |
> > | SCMD          | 0.2352   | 0.2380   | 0.2438   | 0.2528   |
> > | SCMD After    | 0.2614   | 0.2615   | 0.2757   | 0.2795   |
> >
> > **Q3. Where should the line be drawn between hard-to-learn samples and plain outliers?**
> >
> > We appreciate the insightful question. In our method, we employ a percentile-based approach (defaulting to 75th percentile, i.e. $\tau = 0.25$) to distinguish hard-to-learn samples. Specifically, samples with loss exceeding this threshold are selected as hard-to-learn. It is statistically improbable for a quarter of the dataset to consist solely of outliers. In this way, we minimize the risk that these samples are exclusive outliers.
> >
> > To further validate this, we have conducted an analysis, the results of which are detailed in the accompanying tables and in the appendix. Notably, our analysis includes an examination of the model's performance on PACS upon the exclusion of top hard-to-learn samples (specifically, the top 1, top 2, etc., samples with the highest loss).
> >
> > Our results indicate that the removal of these high-loss samples results in a decrease in model performance, which suggests that while these samples are challenging for the model, they are unlikely mere noise and appear to contribute information for the learning process.
> >
> > | Exclude Top-K | Domain A | Domain C | Domain P | Domain S | Average |
> > |-----------|----------|----------|----------|----------|---------|
> > | 0         | 93.0     | 86.8     | 99.0     | 81.1     | 90.0    |
> > | 1         | 92.5     | 86.8     | 99.2     | 80.8     | 89.8    |
> > | 2         | 92.1     | 86.0     | 99.3     | 80.3     | 89.4    |
> > | 3         | 93.0     | 86.4     | 99.3     | 80.6     | 89.8    |
> > | 4         | 92.6     | 86.4     | 98.8     | 80.8     | 89.6    |
> > | 5         | 91.7     | 85.2     | 99.1     | 79.7     | 88.9    |
> >
> > **Q4. The capital letter $P$ is used for both the linear projection and for the data distribution $P_{(X,Y)}$ (when in boldface).**
> >
> > We are thankful for the reviewer's meticulous attention to the notation used in our manuscript. We have revised the manuscript accordingly and have replaced linear projection with $W$.

---

### Review · Reviewer_9y2v · 2024-01-29

**Summary Of Contributions:**

The authors studied a knowledge distillation method for domain generalization leveraging cross modal foundation model, CLIP. This paper proposes a new distillation pipeline equipped with two new modules: hard sample mining and cross-modal module. In addition to the cross-entropy loss, distillation using CLIP-style prediction by the innerproduct between student’s image features and CLIP text features is a different additional component. Lastly, the authors theoretically analyzed the proposed method.

**Audience:**

Yes

**Broader Impact Concerns:**

None.

**Claims And Evidence:**

Yes

**Requested Changes:**

1. Comparison of computational overhead in training / training time
2. Sensitivity analysis for hyperparameters (e.g., sample selection threshold)
3. Table 2, SCMD (final version) has 0 variance but other baselines have some variance. Could you add discussion about this?
4. Table 5, the orders of methods are confusing. Could you place the proposed method on top or at the bottom?

**Strengths And Weaknesses:**

**Strengths**

1. The knowledge distillation with a multi-modal foundation model is a popular idea that enables zero-shot visual recognition including image classification, object detection, semantic segmentation, and visual grounding. In many use cases, the users do not have sufficient computing resources, studying efficient and effective knowledge distillation methods is a right research direction.
2. Knowledge distillation via two classification heads (student’s classifier and CLIP-style classifier with text encoder). This is also an interesting construction.
3. The authors claimed that hard sample mining is effective for knowledge distillation.

**Weaknesses**

1. Heuristics in training scheme. Focusing important samples (e.g., hard sample mining) usually can be more elegantly achieved by new loss functions. For instance, hard negative mining has been actively studied in the context of object detection but dense object detectors often use the focal loss, which is properly modified to address imbalance problem.
2. Efficiency and complex pipeline. The pipeline seems complex and no analysis of computational cost is provided. Filtering/selecting hard samples may require additional neural network inferences and the step may make the pipeline less GPU-friendly. Training time comparison will be useful for readers.

---

> ### Author Response · Authors · 2024-02-10
> **Response to Reviewer 1 (Part 1)**
>
> We sincerely appreciate the reviewer's recognition of the utility of our direction and the interest in our approach. We have addressed the concerns and implemented the suggested changes as detailed below:
>
> **Q1. Comparison of computational overhead in training / inference time.**
>
> We acknowledge the inquiry about the computational load introduced by the feature projectors in SCMD. It is important to note that these modules are utilized exclusively during the training phase and are subsequently discarded, ensuring that they impose no additional burden during inference.
>
> In terms of training overhead, we have added a detailed plot in the appendix of the updated manuscript, comparing the training times of Vanilla KD, ERM, and our SCMD at different phases.
>
> Additionally, the table provided as reference illustrates the average training time per step across four domains in the PACS dataset, with each domain serving as a test domain in turn. SCMD Before and SCMD After indicate the average training time per step for SCMD before and after the transition to full-batch training, respectively. SCMD represents the overall average.
>
> The results indicate that the average training overhead of SCMD is comparable to that of Vanilla KD. This is attributed to our training strategy that focuses on hard-to-learn samples for the majority of the duration, transitioning to full-batch training only in the last $k$ steps. This method balances the overall training overhead.
>
> | Model         | Domain A (sec) | Domain C (sec) | Domain P (sec) | Domain S (sec) |
> |---------------|----------|----------|----------|----------|
> | ERM           | 0.1084   | 0.1082   | 0.1080   | 0.1082   |
> | Vanilla KD    | 0.2431   | 0.2342   | 0.2384   | 0.2435   |
> | SCMD Before   | 0.2264   | 0.2301   | 0.2332   | 0.2438   |
> | SCMD          | 0.2352   | 0.2380   | 0.2438   | 0.2528   |
> | SCMD After    | 0.2614   | 0.2615   | 0.2757   | 0.2795   |
>
>
> **Q2. Sensitivity Analysis for Hyperparameters**
>
> Thank you for addressing this concern. In response to the request, we have conducted an analysis, the results of which are added in the appendix and the accompanying table. To reiterate the definition of the parameters for clarity: for implementation purposes, we set $\tau$ to be the percentile of samples selected as hard-to-learn.
>
> The table presents the performance of our model across four domains of PACS as we adjust the values of $k$ and $\tau$, maintaining all other parameters constant during the experiment.
>
> *$k$ Sensitivity (with $\tau = 0.2$)*: The model's performance remains relatively stable across an array of $k$ values, as evidenced by the average accuracy consistently around 90.0\%.
>
> *$\tau$ Sensitivity (with $k = 0.25$)*: Having very few samples is not helpful to the learning process. And a slight decrement in the model's performance is observed as the value of $\tau$ increases, after reaching 0.2. This trend can be ascribed to the dilution of the impact of hard-to-learn samples as their proportion increases. Specifically, as more samples are classified as hard-to-learn (with a higher $\tau$), the advantage of focusing on these challenging samples is somewhat mitigated, subtly influencing the overall model performance.
>
> | Parameter | Domain A | Domain C | Domain P | Domain S | Average |
> |-----------|----------|----------|----------|----------|---------|
> | **Parameter $k$ (with $\tau = 0.2$)** | | | | | |
> | 0.05      | 93.0     | 87.1     | 99.0     | 81.0     | 90.0    |
> | 0.1       | 93.0     | 87.3     | 99.0     | 81.0     | 90.1    |
> | 0.15      | 93.0     | 86.8     | 99.0     | 80.9     | 89.9    |
> | 0.2       | 93.0     | 87.2     | 99.0     | 80.9     | 89.9    |
> | 0.25      | 93.0     | 86.8     | 99.0     | 86.9     | 90.0    |
> | 0.3       | 93.0     | 86.9     | 99.0     | 80.7     | 89.9    |
> | 0.35      | 93.0     | 86.9     | 99.0     | 81.0     | 90.0    |
> | 0.4       | 92.3     | 87.0     | 98.9     | 81.6     | 89.9    |
> | 0.45      | 92.3     | 86.9     | 99.0     | 81.3     | 89.9    |
> | 0.5       | 92.3     | 86.1     | 99.0     | 81.2     | 89.6    |
> | **Parameter $\tau$ with ($k = 0.25$)** | | | | | |
> | 0.05      | 91.4     | 85.1     | 98.9     | 81.5     | 89.2    |
> | 0.1       | 92.4     | 85.3     | 99.1     | 81.3     | 89.5    |
> | 0.15      | 92.6     | 86.9     | 99.0     | 80.6     | 89.8    |
> | 0.2       | 93.0     | 86.8     | 99.0     | 81.1     | 90.0    |
> | 0.25      | 92.1     | 86.6     | 99.3     | 81.0     | 89.7    |
> | 0.3       | 93.0     | 86.1     | 99.4     | 79.7     | 89.6    |
> | 0.35      | 92.7     | 87.2     | 99.5     | 78.4     | 89.4    |
> | 0.4       | 93.1     | 87.2     | 99.5     | 78.6     | 89.6    |
> | 0.45      | 93.3     | 86.7     | 99.3     | 79.0     | 89.6    |
> | 0.5       | 92.2     | 86.7     | 99.2     | 79.0     | 89.3    |

---

> > ### Author Response · Authors · 2024-02-10
> > **Response to Reviewer 1 (Part 2)**
> >
> > **Q3. SCMD in table 2 has 0 variance**
> >
> > We appreciate the reviewer's attention to the 0 variance observed for SCMD in Table 2. We have checked our experimental procedures carefully and we do not believe it has issues. It is mostly likely because the results indeed (coindentally) come out with 0 variance.
> >
> >
> > **Q4. The orders of methods are confusing**
> >
> > We appreciate your attention to the organization of the methods in the table. Following your feedback, we have promptly revised the table.
> >
> > **Q5. New loss functions**
> >
> > In response to the suggestion, we have explored the integration of focal loss, specifically by substituting the CE loss with a focal loss. The outcomes are added to Table 2. Specifically, when the model trains and selects samples based on focal loss, the performance on the PACS is observed to the $89.2 \pm 0.4$, which is lower compared to the $90.1 \pm 0.0$ performance achieved when using CE loss.

---

### Author Response · Authors · 2024-02-10
**Paper Revision**

We appreciate the insightful comments from the reviewers. Based on the feedback, we have modified our submission. The major updates are as follows:

* We have updated the table by placing our proposed method at the bottom for clarity and have added a comparison with focal loss in Table 2.

* We have included plots and analyses for hyperparameter sensitivity, computational time, and analysis of potential outliers.

---

### Decision · Action_Editor_32pR · 2024-03-17

**Recommendation:** Accept as is

**Comment:**

This paper shows an effective CLIP-based domain generalization:

-It introduces a novel distillation pipeline leveraging the CLIP model, showcasing a new approach to domain generalization that harnesses cross-modal capabilities.
-With the introduction of hard sample mining and a cross-modal module, the paper contributes new methods to the knowledge distillation field, evidenced by performance improvements on benchmark tasks.
-Comprehensive ablation study validates the effectiveness of each proposed component, providing clear evidence that they contribute to the model's enhanced performance.

Initially, the reviewers raised concerns about insufficient comparative experiments, unaddressed computation efficiency, and hyperparameter selection. After the response, the authors have addressed all the concerns. All reviewers recommend acceptance.

**Audience:**

Domain generalization in the large model era is still worth studying, and the way how we study it should be changed. This paper shows a promising direction.

**Claims And Evidence:**

Use of CLIP for Domain Generalization: The submission leverages a pre-trained CLIP model to propose a novel knowledge distillation pipeline for domain generalization. The integration of hard sample mining and a cross-modal module is an innovative approach. The evidence supporting the effectiveness of this method includes performance improvements on the DomainBed benchmark, demonstrating the potential of leveraging CLIP's capabilities for domain generalization tasks.

Effectiveness of the Proposed Method SCMD: The ablation study, which shows that each proposed component contributes to the performance boost, provides direct evidence supporting the claim that the methodological innovations introduced in the paper (hard-to-learn sample selection and cross-modality module) are effective.